# In Silico Identification of Dysregulated miRNAs Targeting *KRAS* Gene in Pancreatic Cancer

**DOI:** 10.3390/diseases12070152

**Published:** 2024-07-12

**Authors:** Asbiel Felipe Garibaldi-Ríos, Luis E. Figuera, Guillermo Moisés Zúñiga-González, Belinda Claudia Gómez-Meda, Patricia Montserrat García-Verdín, Irving Alejandro Carrillo-Dávila, Itzae Adonai Gutiérrez-Hurtado, Blanca Miriam Torres-Mendoza, Martha Patricia Gallegos-Arreola

**Affiliations:** 1División de Genética, Centro de Investigación Biomédica de Occidente (CIBO), Centro Médico Nacional de Occidente (CMNO), Instituto Mexicano del Seguro Social (IMSS), Guadalajara 44340, Jalisco, Mexico; asbiel.garibaldi4757@alumnos.udg.mx (A.F.G.-R.); luisfiguera@yahoo.com (L.E.F.); patricia.garcia@alumnos.udg.mx (P.M.G.-V.); irving.carrillo4754@alumnos.udg.mx (I.A.C.-D.); 2Doctorado en Genética Humana, Centro Universitario de Ciencias de la Salud (CUCS), Universidad de Guadalajara (UdeG), Guadalajara 44340, Jalisco, Mexico; 3División de Medicina Molecular, Centro de Investigación Biomédica de Occidente (CIBO), Centro Médico Nacional de Occidente (CMNO), Instituto Mexicano del Seguro Social (IMSS), Sierra Mojada 800, Col. Independencia, Guadalajara 44340, Jalisco, Mexico; mutagenesis95@hotmail.com; 4Instituto de Genética Humana “Dr. Enrique Corona Rivera”, Departamento de Biología Molecular y Genómica, Centro Universitario de Ciencias de la Salud (CUCS), Universidad de Guadalajara (UdeG), Guadalajara 44340, Jalisco, Mexico; belinda.gomez@academicos.udg.mx (B.C.G.-M.); itzae.gutierrez@academicos.udg.mx (I.A.G.-H.); 5Laboratorio de Inmunodeficiencias Humanas y Retrovirus, División de Neurociencias, Centro de Investigación Biomédica de Occidente (CIBO), Centro Médico Nacional de Occidente (CMNO), Instituto Mexicano del Seguro Social (IMSS), Guadalajara 44340, Jalisco, Mexico; blanca.torresm@imss.gob.mx; 6Departamento de Disciplinas Filosófico-Metodológicas, Centro Universitario de Ciencias de la Salud (CUCS), Universidad de Guadalajara (UdeG), Guadalajara 44340, Jalisco, Mexico

**Keywords:** pancreatic neoplasms, *KRAS* proto-oncogene, microRNAs, biomarkers, in silico techniques, gene expression regulation

## Abstract

Pancreatic cancer (PC) is highly lethal, with *KRAS* mutations in up to 95% of cases. miRNAs inversely correlate with *KRAS* expression, indicating potential as biomarkers. This study identified miRNAs targeting *KRAS* and their impact on PC characteristics using in silico methods. dbDEMC identified dysregulated miRNAs in PC; TargetScan, miRDB, and PolymiRTS 3.0 identified miRNAs specific for the *KRAS* gene; and OncomiR evaluated the association of miRNAs with clinical characteristics and survival in PC. The correlation between miRNAs and *KRAS* was analysed using ENCORI/starBase. A total of 210 deregulated miRNAs were identified in PC (116 overexpressed and 94 underexpressed). In total, 16 of them were involved in the regulation of *KRAS* expression and 9 of these (hsa-miR-222-3p, hsa-miR-30a-5p, hsa-miR-30b-5p, hsa-miR-30e-5p, hsa-miR-377-3p, hsa-miR-495-3p, hsa-miR-654-3p, hsa-miR-877-5p and hsa-miR-885-5p) were associated with the clinical characteristics of the PC. Specifically, the overexpression of hsa-miR-30a-5p was associated with PC mortality, and hsa-miR-30b-5p, hsa-miR-377-3p, hsa-miR-495-3p, and hsa-miR-885-5p were associated with survival. Correlation analysis revealed that the expression of 10 miRNAs is correlated with *KRAS* expression. The dysregulated miRNAs identified in PC may regulate *KRAS* and some are associated with clinically relevant features, highlighting their potential as biomarkers and therapeutic targets in PC treatment. However, experimental validation is required for confirmation.

## 1. Introduction

Pancreatic cancer (PC) is one of the most aggressive cancers and a leading cause of death worldwide [1,2]. According to the most recent data published by the Global Cancer Observatory (GLOBOCAN), in 2022, 510,992 new cases of the disease were diagnosed and 467,409 deaths were reported [3]. Moreover, it is expected that the incidence and mortality of this disease will increase, surpassing other neoplasms, including colon cancer, in the coming years [4].

The etiologic factors contributing to the development of this neoplasm are multifactorial. Lifestyle-related risk factors include alcohol and tobacco consumption as well as obesity and diabetes [1], while the genetic predisposition to PC is undeniable. Pathogenic variants and germline mutations in genes, such as *KRAS*, *TP53*, and *CDKN2A*, among others, play significant roles [5]. However, a key feature of PC is mutations in several oncogenes. Among these, *KRAS*, a member of the RAS oncogene family, is mutated in up to 95% of PC cases [6,7].

The *KRAS* gene encodes the *KRAS* protein, a key player in a variety of cellular activities, functioning as a switch in cellular signal transduction. In its inactive state, *KRAS* remains bound to guanosine diphosphate (GDP). However, upon receiving a mitotic signal upstream, several guanine nucleotide exchange proteins (GEFs), such as SOS and GRB2, facilitate *KRAS* activation and guanosine triphosphate (GTP) loading. When bound to GTP, *KRAS* undergoes conformational changes in regions known as “SWITCH I and II”, adopting an open and active conformation and binding to its effector proteins. This cascade transduces signals that lead to cell proliferation, differentiation, metabolism, and migration [8,9,10,11,12].

Approximately 95% of oncogenic *KRAS* mutations are changes located in codons 12 (G12), 13 (G13), or 61 (Q61), which are precisely located in the Switch I and II domains and lead to increased *KRAS* activity and, consequently, to the continued activation of effector proteins [13]. Even the KRASG12D mutation is associated with poor patient prognosis [7].

According to the cBioPortal platform [14], in PC, the *KRAS* gene has 119 types of missense mutations, which can affect both gene expression and the clinical features of the disease. Furthermore, according to GEPIA [15], statistically significant differences in *KRAS* expression are observed between PC patients and healthy individuals, with expressions of 3.79 TPM (transcripts per million) and 2.14 TPM, respectively.

Furthermore, GEPIA shows that the group with high *KRAS* gene expression has an increased risk of death compared to the low expression group (Logrank *p* = 0.013; HR(high) = 1.7; *p*(HR) = 0.014; n(high) = 88; n(low) = 88). It has also been suggested that high *KRAS* expression is associated with worse disease-free survival compared to low *KRAS* expression (Logrank *p* = 0.0065; HR(high) = 1.8; *p*(HR) = 0.0076; n(high) = 88; n(low) = 88).

Although the role of *KRAS* and its point mutations in PC have been well studied, recent evidence suggests that *KRAS*, due to its large 3′UTR region, may be significantly regulated by miRNAs [16]. miRNAs are short (18–25nt) single-stranded RNAs that associate with other proteins, such as AGO2, to form the RNA-induced silencing complex (RISC). This complex regulates gene expression post-transcriptionally by binding in a complementary manner to the 3′UTR region of target genes, thereby modulating their expression [17,18].

Several studies have elucidated the role of miRNAs in *KRAS* regulation, including those belonging to the lethal 7 (Let-7) family. Comprehensive analyses of miRNA expression profiles across various tumour types have been performed and compared with *KRAS* expression. The results indicate that certain miRNAs exhibit a significant negative correlation with *KRAS* expression. Moreover, in vitro models have demonstrated that the overexpression of these miRNAs leads to a significant decrease in *KRAS* levels. These findings suggest the direct regulation of *KRAS* by these miRNAs [19,20].

It has recently been proposed that oncogenic *KRAS* promotes the dysregulation of miRNAs by promoting the formation of “*KRAS*-induced granules” (KG), which are composed of accumulations of pro-tumorigenic transcripts caused by the global dysregulation of miRNAs [21]; this occurs due to the suppression of several proteins, leading to a reduction in AGO2 activity [22].

Additionally, reports suggest that *KRAS* can be indirectly regulated by miRNAs, such as through the modulation of effector genes or *KRAS* regulators [19].

Although it is well known that *KRAS* is widely regulated by miRNAs, there are currently no studies that reveal the global expression of miRNAs regulating *KRAS* in PC. In this study, dysregulated miRNAs in PC and their binding or interaction with *KRAS* were analysed in silico.

## 2. Materials and Methods

### 2.1. Differential Expression of miRNAs in PCs

The expression profiles of dysregulated miRNAs in PC were analysed using the dbDEMC (Database of Differentially Expressed MiRNAs in Human Cancers) platform (https://www.biosino.org/dbDEMC/index [accessed on 5 January 2024] [23]. dbDEMC serves as a comprehensive data repository that aggregates information on miRNA expression across diverse human neoplasms. This tool integrates expression data from repositories such as GEO (Gene Expression Omnibus), Sequence Read Archive (SRA), ArrayExpress, and The Cancer Genome Atlas (TCGA). For the analysis of the miRNA expression profiles in PC, the experiment “EXP00134” (PMID: 21953293) [24] was used, whose source is the GEO platform. In this experiment, the expression profiles of PC cells were compared with those of normal pancreatic ductal cells (NPDCs), which were used as controls. The instrument used for this experiment was the Agilent-019118 Human miRNA Microarray (Agilent technologies, Santa Clara, CA, USA).

microRNAs with a Log Fold Change (LogFC) of ±0.05, along with significant *p*-values < 0.05 and adjusted *p*-values (*p*.adj) < 0.05, were considered dysregulated, categorized as overexpressed (up) or underexpressed (down).

### 2.2. Identification of KRAS-Targeted miRNAs

To identify miRNAs dysregulated in PC with potential regulatory effects on the *KRAS* gene through complementarity, we used TargetScan (https://www.targetscan.org/vert_80/ [accessed on 6 January 2024]) [25], miRDB (https://mirdb.org [accessed on 6 January 2024]) [26], and PolymiRTS 3.0 (https://compbio.uthsc.edu/miRSNP/search.php [accessed on 6 January 2024]) [27]. In addition, the last platform was used to determine the precise location in the gene where these miRNAs could bind, based on their complementarity with *KRAS*.

TargetScan uses a variety of algorithms to identify base complementarity between the seed region of miRNAs and the 3′UTR regions of mRNAs. On the other hand, miRDB relies on the results of high-throughput sequencing experiments to predict miRNA targets. PolymiRTS stores numerous genetic variations, both in the seed region of miRNAs and in the 3′UTR region of the targeted genes, thus allowing, through various algorithms and genomic alignments, the comparison and prediction of miRNA-mRNA target sites.

### 2.3. Differential Expression of KRAS-Targeted miRNAs in PCs

After identifying the dysregulated miRNAs in PC targeting the *KRAS* gene, their expression profiles were compared. To accomplish this, the dbDEMC meta-profiling tool was utilised, which integrates expression data from multiple experiments and assesses changes in logFC. Subsequently, to ascertain the expression profiles of each miRNA in the comparison between PC and NPDC, the same tool was employed, utilising the EXP00134 [24] experiment. Dysregulated miRNAs were determined based on logFC with *p*-values < 0.05 and *p*.adj < 0.05.

The OncomiR tool (https://oncomir.org [accessed on 8 January 2024]) [28] was used to examine how the expression profiles of each *KRAS*-targeted miRNA correlated with various clinical features of PC, as well as to compare the expression levels to patient survival or mortality.

OncomiR is a comprehensive platform that uses TCGA data to perform statistical analyses and detect dysregulated microRNAs in human neoplasms and explore their association with the clinicopathologic features of these diseases.

To investigate the association between miRNA expression and the clinical features of PC, as well as its impact on patient survival or mortality, statistical thresholds were established with a significance level of *p* < 0.05.

### 2.4. miRNA- KRAS mRNA Co-Expression Analysis

Utilising the ENCORI (Encyclopedia of RNA Interactomes) tool/StarBase (https://rnasysu.com/encori/ [accessed on 8 January 2024]) [29], a co-expression analysis was conducted to elucidate the interplay between miRNA and *KRAS* mRNA expression. ENCORI/StarBase is a platform that integrates high-throughput sequencing data to facilitate such analyses.

The data were evaluated on a logarithmic scale (log2), and a statistical significance threshold of *p* < 0.05 was set.

## 3. Results

### 3.1. Differential Expression of miRNAs in PC

An analysis of the EXP00134 [24] experiment using dbDEMC revealed the dysregulation of 210 miRNAs in PC, with 116 upregulated and 94 downregulated miRNAs (Figure 1) (Appendix A).

### 3.2. Identification of KRAS-Targeted miRNAs

Using the TargetScan and miRDB tools, it was identified that out of the 210 dysregulated miRNAs in PC, 16 of them specifically targeted the *KRAS* gene. A further analysis with PolyMIRTS revealed that these miRNAs interacted with *KRAS* in regions harbouring several single nucleotide variants (SNVs) (Table 1).

### 3.3. Differential Expression of KRAS-Targeting miRNAs in PC

A meta-profiling analysis conducted in dbDEMC revealed the significant dysregulation of miRNAs targeting the *KRAS* gene in PC. Changes in the expression levels (logFC) were observed with a range varying from 5 to −10 (Figure 2A).

In the differential expression analysis of miRNAs from experiment EXP00134 [24], these miRNAs were found to be downregulated in PC compared to NPDC, with a logFC ranging from −5.18 (hsa-miR 217; *p* = 0.00000173; *p*.adj = 0.0000385) to 1.71 (hsa-miR-877-5p; *p* = 0.0164, *p*.adj = 0.0443) with *p* and *p*.adj < 0.005 (Figure 2B, Appendix A).

Additionally, the experimental validation data for these miRNAs were analysed on the same platform. It was observed that the deregulation of hsa-miR-885-5p, hsa-miR-326, hsa-miR-30b-5p, hsa-miR-30a-5p, hsa-miR-222-3p, and hsa-miR-217 has been experimentally validated in several studies that investigated their expression in different types of cancer (Table 2).

On the OncomiR platform, the expression of the miRNAs hsa-miR-222-3p, hsa-miR-30a-5p, hsa-miR-30b-5p, hsa-miR-30e-5p, hsa-miR-377-3p, hsa-miR-495-3p, hsa-miR-654-3p, hsa-miR-877-5p and hsa-miR-885-5p was found to be associated with the clinical features of PC (Figure 3, Appendix A).

In addition, significant differences were observed between the expression of several miRNAs and the clinical outcomes in terms of patient mortality and survival.

The overexpression of hsa-miR-30a-5p was observed to be higher in deceased patients, whereas the expression of hsa-miR-30b-5p, hsa-miR-377-3p, hsa-miR-495-3p, and hsa-miR-885-5p was higher in disease survivors (Table 3).

### 3.4. miRNA- KRAS mRNA Co-Expression Analysis

The co-expression profiles of miRNAs and *KRAS* mRNA were examined, revealing a significant correlation between the expression profiles of six miRNAs and *KRAS* mRNA. Notably, a positive correlation was observed between hsa-miR-222-3p and *KRAS* mRNA (r = 0.467, *p* = 5.21× 10^−11^), as well as between hsa-miR-34b-3p and *KRAS* mRNA (r = 0.281, *p* = 1.44× 10^−4^) (Figure 4).

### 3.5. Molecular Interaction

Using the above observations obtained from the in silico data, we can infer the miRNA-mediated regulation pathway of *KRAS* and its role in PC (Figure 5).

## 4. Discussion

Currently, PC remains one of the most deadly cancers worldwide [2]. In addition, the genetic factors of this pathology are undeniable, as the *KRAS* gene is mutated in up to 95% of cases of this pathology [6]. It has been shown that *KRAS* has a remarkably large 3′UTR region, suggesting that its regulation is post-transcriptionally mediated by several miRNAs [43]. Therefore, the study of this miRNA-*KRAS* regulation could lead to the identification of miRNAs useful as molecular markers for early diagnosis, disease prognosis, and the development of more effective, targeted therapies. Recently, miRNAs have gained significant importance in cancer therapies due to their ability to regulate the gene expression involved in tumour progression and metastasis. miRNA-based therapeutic strategies include the use of miRNA mimetics to restore the function of tumour suppressor miRNAs and antagomirs to inhibit the action of oncogenic miRNAs [44].

The present study computationally identified the expression profiles of miRNAs in PC, as well as their relationship with *KRAS*.

In our study, we identified 210 miRNAs dysregulated in PC, of which 116 were overexpressed and 94 were underexpressed. Of these 210 miRNAs dysregulated in the disease, with predictions performed in TargetScan, miRDB, and PolymiRTS 3.0, 16 were found to target specific loci in the 3′UTR region of *KRAS*, regions that are also polymorphic. In this sense, it has been reported that variants in the 3′UTR region of the *KRAS* gene could modify the miRNA-binding sites that could regulate it [19,44,45], and that some of these variants have also been associated with different types of neoplasms [46,47,48,49,50,51].

One of the miRNAs dysregulated in PC and targeting *KRAS* is hsa-miR-885-5p, which has previously been reported [52] as a tumour suppressor in hepatocellular carcinoma by regulating AEG1 and as an inhibitor of invasion and metastasis in gastric cancer by regulating ME1 [53]. Furthermore, in this study, we observed that, in *KRAS*, this miRNA targets the rs8720 locus, an SNV that has previously been associated with colorectal cancer [49,54].

Furthermore, we observed that hsa-miR-885-5p is overexpressed in PC with a logFC of 1.41, and it has been experimentally demonstrated that it is also overexpressed in pheochromocytoma [30]. In OncomiR, we observed that the expression of this miRNA is also significantly correlated with the histologic grade, pathologic N stage, and pathologic T stage of PC, and its overexpression is also significantly associated with patient survival. Thus, in addition to regulating *KRAS*, *AEG1* and *ME1*, this miRNA might coordinate PC progression by regulating other important genes in this pathology. Furthermore, a negative correlation between the expression of this miRNA and *KRAS* was observed in the co-expression analysis, suggesting that when miRNA expression increases, *KRAS* expression decreases. These findings could indicate that this miRNA may play a critical role as a prognostic marker of disease progression and survival by regulating the expression of *KRAS* or other oncogenes, thereby inhibiting cell invasion and impeding cancer progression.

Another miRNA dysregulated in PC and targeting *KRAS* is hsa-miR-877-5p, which has previously been reported as a marker of bone metastasis in lung cancer [55] and is overexpressed in hepatocellular carcinoma [56]. In this study, we observed that this miRNA targets the rs712 region of *KRAS*, an SNV that has been associated with colorectal cancer [57], breast cancer [58], and breast cancer metastasis [59]. Therefore, this variant could affect the regulation of *KRAS* by miRNAs and could possibly promote the neoplastic process. This suggests that this variant could influence the regulation of *KRAS* by miRNAs and potentially promote the neoplastic process. hsa-miR-877-5p was also found to be overexpressed in PC (logFC 1.71), which is consistent with previously reported studies [56], and that it is associated with the pathological state of the disease.

We found that hsa-miR-654-3p is underexpressed in PC with a logFC −1.59, and that it is also associated with the histologic grade and pathologic status of the disease, so this miRNA may regulate other genes that are also important in the progression of PC. 

On the other hand, hsa-miR-647, which was found to be overexpressed in PC patients (logFC 1.49), has also been reported as an inhibitor of hepatocellular carcinoma progression by possibly regulating *PTPRF* [60], as a regulator of glioma progression and invasion by directly regulating *HOXA9* [61], and conversely as a tumour promoter in gastric cancer by regulating the *TP73* gene [62]. This suggests that miRNAs may play a critical role as regulators of tumour suppressor genes or oncogenes, influencing the suppression or promotion of carcinogenesis by modulating the expression of key genes in these diseases.

Another miRNA found to be underexpressed in PC (logFC-1.33) is hsa-miR-548c-5p, which has previously been associated with disease-free survival in breast cancer [63]. The above suggests that the silencing of this miRNA may facilitate the overexpression or dysregulation of critical genes in tumour progression.

hsa-miR-513b-5p was also found to be overexpressed in PC (logFC, 1.47), and similar data were found in a previous study [64] where the overexpression of this miRNA suppressed testicular embryonal carcinoma cell proliferation in vitro and induced apoptosis. On the other hand, the silencing of this miRNA reversed this effect by upregulating *IRF2* expression. Therefore, it was observed that the overexpression of this miRNA could induce *IRF2* silencing and, subsequently, *TP53* expression. This could indicate that this miRNA could interact in various cell signalling networks, which could in turn influence cell growth.

In this study, hsa-miR-495-3p was found to be underexpressed in PC with a logFC of −2.13; in a previous study, it was shown that this molecule can regulate *TWIST1* and inhibit metastasis in gastric cancer [65]. Furthermore, it has been reported that the loss of expression in this miRNA could induce the overexpression of epigenetic modifiers, which could in turn modify the expression of key genes in malignancy and the growth of gastric epithelial cells [66]. This is in line with the observations made in this study, where it was shown in silico that the expression of this miRNA is associated with the key features of PC, such as the histologic grade and pathologic state, mainly M and T pathologic states. Furthermore, in this study, the overexpression of this miRNA was associated with PC survival and was negatively correlated with *KRAS* expression, suggesting that the downregulation of this miRNA may have a significant effect on the overexpression of genes that regulate the tumour characteristics of PC.

A previous study reported that the expression levels of the miRNA hsa-miR-377-3p is lower in melanoma patients and that its overexpression inhibits cell growth by directly regulating *ARMC8* [67], which is inconsistent with our findings, as we found an underexpression of this miRNA in patients with PC (logFC −2.39). Furthermore, we observed that this miRNA is associated with the pathological T status and its overexpression with disease survival. This suggests that the underexpression of this miRNA could enable the expression of genes such as *KRAS* or other oncogenes that promote cellular proliferation and differentiation. In the case of PC, these genes are implicated in the pathological state of the disease. Therefore, if this miRNA is overexpressed, it might suppress the effect of these genes, leading to an increase in the survival of individuals with this pathology.

hsa-miR-34b-3p was found to be overexpressed in PC with a logFC of 1.47, which is consistent with a previous study [68], where this miRNA was also found to be overexpressed in oesophageal squamous cell carcinoma. Additionally, in our study, the co-expression of this miRNA was found to be positively correlated with *KRAS* expression and could regulate the gene in the genomic regions of the SNV rs61764370 variant, which was previously associated with breast cancer in the Mexican population [69] and lung cancer in a population from Iran [70]. The overexpression of this miRNA in PC and oesophageal squamous cell carcinoma might suggest that this miRNA could also regulate and reduce the expression of tumour suppressor genes. Furthermore, the analysis of the association of *KRAS* SNVs where this miRNA binds could indicate that subtle changes, such as an SNV in the miRNA-binding sites in the 3’UTR region, might influence the regulation of *KRAS* by miRNAs. This underscores the intricate regulatory mechanisms involved in cancer development and highlights the potential importance of targeting these regulatory pathways for therapeutic intervention.

In a previous study [71], hsa-miR-326 was found to inhibit the development of non-small cell lung cancer by directly regulating the *CCND1* gene. It was also reported that this miRNA is underexpressed in this pathology, which is consistent with the results obtained in this analysis, where it has been experimentally proven that this miRNA is expressed in medulloblastoma [31] and Barrett’s carcinogenesis [32]. However, it contrasts with the results obtained from PC, where this miRNA was observed to be overexpressed (logFC 1.69). Furthermore, it was observed that the expression of this miRNA positively correlates with the expression of *KRAS*, so this miRNA could play a role in tumour promotion in PC, somehow repressing the expression of genes that suppress cell growth in PC.

In this study, we observed that hsa-miR-30e-5p is downregulated in PC (logFC −0.68) and that its expression is associated with the pathological status T of the disease, which is similar to previously reported results [72], where it was identified that this miRNA participates in regulatory pathways that confer radiation resistance to nasopharyngeal carcinoma cell lines by regulating genes such as *MAPK1*, *SOS1*, *TGFBR1*, *TGFBR2*, *TP53*, *CASP3*, *CCNE2*, *PTEN*, and *CDK2*. In addition, it has been found that this miRNA acts as a tumour suppressor by directly regulating the Cyb561/ROS pathway’s signalling in acute myeloid leukaemia [73], and that this molecule mediates carcinogenic processes in colorectal cancer by directly regulating *ITGA6* and *ITGB* [74] and those in prostate cancer by regulating *CTHRC1* [75]. Many of these genes are closely related to *KRAS*, so it is to be expected that the dysregulation of this miRNA could be involved in PC tumour promotion by suppressing the expression of genes related to cell proliferation or allowing the expression of those that promote tumour progression. In addition to the above-mentioned tumours, this miRNA has also been associated with the occurrence, development and prognosis of several types of cancer including cervical squamous cell carcinoma [76], lung cancer [77], and chronic myeloid leukaemia [78], by directly regulating various genes and molecular pathways in these pathologies.

On the other hand, we observed that hsa-miR-30b-5p is underexpressed in PC (−0.78), and that its expression is associated with the histologic grade and pathologic stage M of the disease. Furthermore, we observed that this molecule has been experimentally reported to be underexpressed in colon cancer [33] and medulloblastoma [31], while this miRNA was reported to be overexpressed in breast cancer [34]; it has also been suggested that by decreasing its expression, it acts as a promoter of metastasis in gastric cancer [79]. This again suggests that this miRNA, like others, can have both tumour suppression and tumour promotion effects, depending on the gene it regulates in each pathology. For example, in this study, it was observed that the overexpression of this miRNA is associated with the survival of PC and that its expression negatively correlates with the expression of *KRAS*, so that, in PC, this miRNA may possibly play a tumour suppressor role by directly regulating this gene.

The miRNA hsa-miR-30a-5p was previously reported to be a tumour suppressor in lung squamous cell carcinoma by directly targeting the *ATG5* gene [80] an ad suppressor of hepatocellular carcinoma tumour migration and invasion by directly regulating *SNAIL1* [81]. In addition, it has been described that the inhibition of this miRNA promotes chemoresistance in lung cancer by directly regulating the expression of *BECN1* [82]. In addition, downregulation has been reported in lung adenocarcinoma [35] and non-small cell lung cancer [83], which is consistent with this study, where the downregulation of this miRNA was observed in PC (logFC −0.82); we also observed that the expression of this molecule is associated with the M and N pathological state and that its overexpression is associated with PC mortality. This could indicate that when this miRNA is downregulated, it may potentially inhibit the expression of oncogenes, whereas when it is overexpressed, it could downregulate the expression of tumour suppressor genes.

We found that hsa-miR-222-3p is overexpressed in PC (logFC 1.53), and the overexpression of this miRNA was previously reported to be associated with the promotion of metastasis in lung cancer [84]. This is consistent with our study, where we observed that this miRNA correlated with the histological grade and pathological N-stage, while its expression correlated positively with *KRAS* expression. This may indicate that the overexpression of this miRNA negatively regulates the expression of genes that suppress *KRAS* activity or the expression of genes that suppress cell proliferation in PC. Furthermore, in this study, we again observed that the deregulation of this miRNA has different effects in different types of tumours; for example, downregulation was experimentally observed in prostate cancer [36], while upregulation was reported in lymphoma [37], cholangiocarcinoma [38], and kidney cancer [40].

Recently, a study reported that hsa-miR-217 expression plays a tumour suppressor role in non-small cell lung cancer cells by targeting *SIRT1* [85]. In this study, we observed that this miRNA is significantly downregulated and underexpressed in PC (logFC −5.18). Furthermore, experimentally, this miRNA has been observed to be underexpressed in PC [41] and overexpressed in lung cancer [42]. In this study, we observed that the expression of this molecule correlates negatively with the expression of *KRAS*. Thus, it fulfils a possible suppressor role in PC tumorigenesis.

In a study [86] aimed at identifying downregulated miRNAs in metastatic colorectal cancer patients resistant to irinotecan-based treatment, hsa-miR-181d-5p was found to be overexpressed in this group of patients. In our study, we observed that this miRNA is overexpressed in PC (logFC 0.76) and that it modulates *KRAS* within the genomic locus of SNV rs9266, which has been previously reported to be associated with breast cancer in the Mexican population [69] and a sample of the Chinese population [48]. This supports that the variant located in the 3′UTR region of *KRAS* could modify the miRNA-binding sites that potentially regulate the gene, thereby promoting the process of carcinogenesis.

Given that *KRAS* can be indirectly regulated by miRNAs through the genes interacting with *KRAS* [19], it is crucial to highlight the potentially pivotal role of miRNAs in tumour promotion in PC and other cancers through complex regulatory networks. For instance, a recent study revealed that miRNAs also regulate *YAP/TAZ* in the Hippo pathway, a critical pathway in diverse cellular activities [87]; the activation of *YAP/TAZ* was also significantly associated with resistance to inhibitors specifically targeting *KRAS*. In addition, the combined inhibition of the Hippo pathways has been shown to improve the response of tumours to treatment with *KRAS* G12C-specific inhibitors [88,89], suggesting significant therapeutic potential in cancer treatment and the integration of miRNAs that not only indirectly regulate *KRAS*, but also all those genes that regulate and are regulated by *KRAS*.

## 5. Conclusions

A total of 210 miRNAs dysregulated in PC were identified, of which 16 could potentially regulate *KRAS*. The miRNAs hsa-miR-222-3p, hsa-miR-30a-5p, hsa-miR-30b-5p, hsa-miR-30e-5p, hsa-miR-377-3p, hsa-miR-495-3p, hsa-miR-654-3p, hsa-miR-877-5p, and hsa-miR-885-5p were associated with the relevant clinical features of the pathology, while the overexpression of hsa-miR-30a-5p was associated with PC mortality, and the overexpression of hsa-miR-30b-5p, hsa-miR-377-3p, hsa-miR-495-3p, and hsa-miR-885-5p was associated with survival. The expression of some miRNAs was found to correlate with *KRAS* expression, mainly the miRNAs hsa-miR-34b-3p and hsa-miR-222-3p. The miRNAs dysregulated in PC target specific loci within the 3′UTR regions of *KRAS*, and some of them (rs8720, rs712, rs190084851, rs61764370 and rs9266) have been previously associated with different types of cancer. The above findings highlight the potential of miRNAs as biomarkers and possible therapeutic targets in the treatment of PC; however, experimental validation is needed to confirm these results.

## Figures and Tables

**Figure 1 diseases-12-00152-f001:**
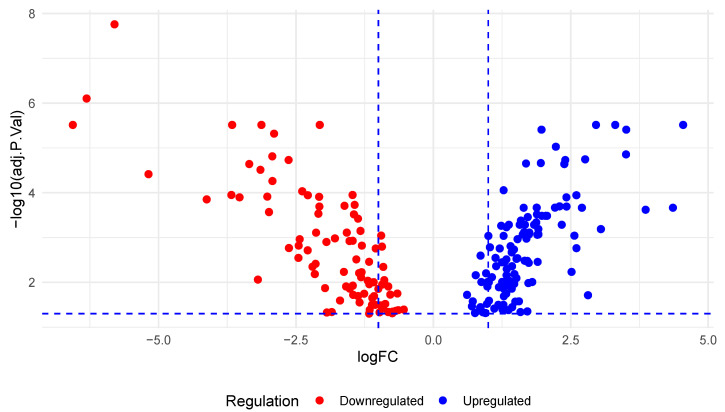
miRNAs dysregulated in PC.

**Figure 2 diseases-12-00152-f002:**
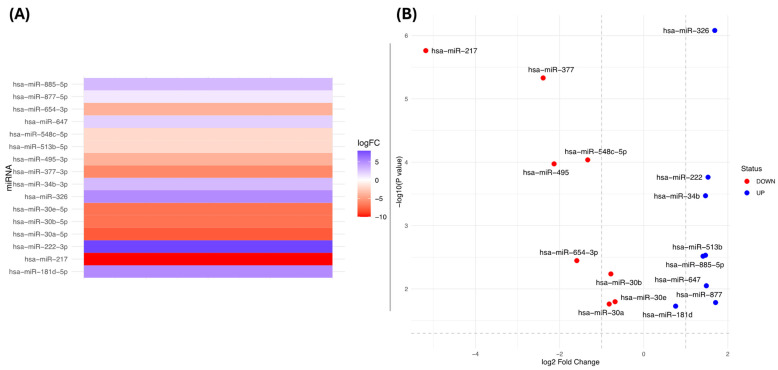
(**A**) Meta-profiling of miRNAs dysregulated in PC and targeting *KRAS*. (**B**) miRNAs dysregulated in PC (EXP00134 [24]) and targeting *KRAS*.

**Figure 3 diseases-12-00152-f003:**
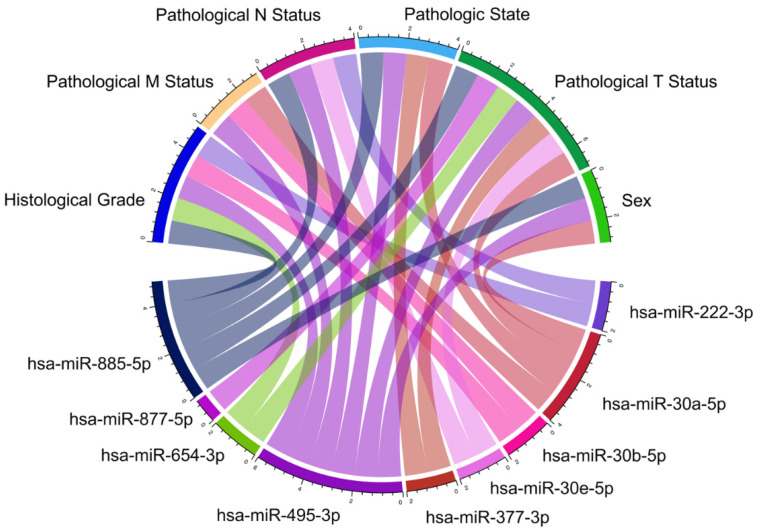
Association between miRNAs and clinical features of PC. As shown in the figure, miRNAs are linked to various clinicopathological features of PC. For instance, the expression of hsa-miR-222-3p is associated with the histologic grade of the disease and pathologic stage N. Similarly, the other miRNAs are also associated with various characteristics.

**Figure 4 diseases-12-00152-f004:**
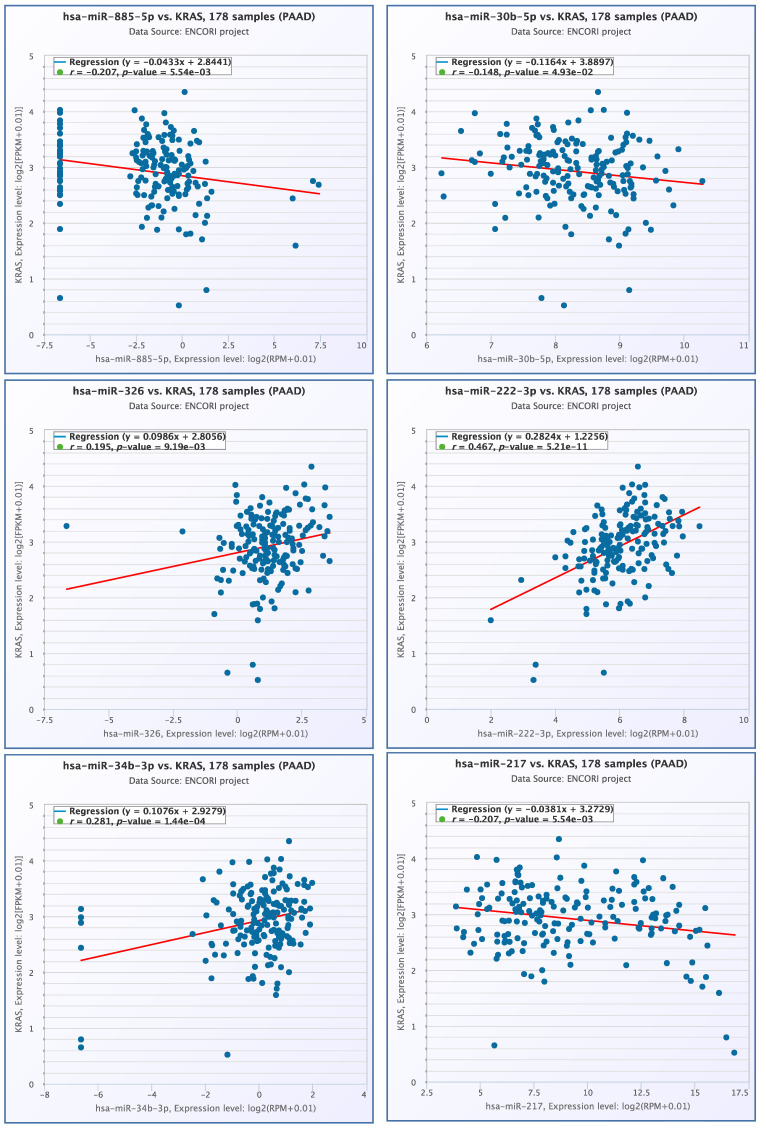
*miRNAs-KRAS* mRNA expression profiles.

**Figure 5 diseases-12-00152-f005:**
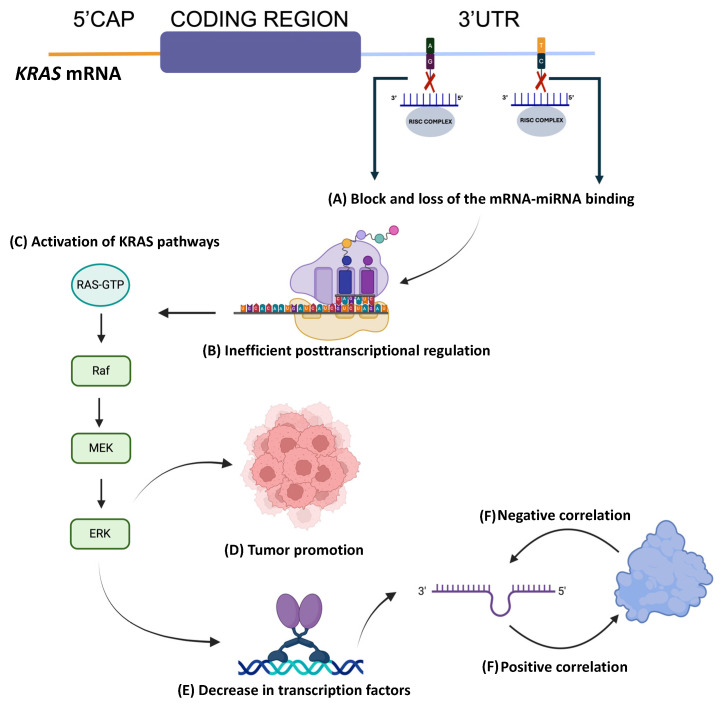
General overview of *KRAS* regulation by miRNAs. (**A**) Variants located at the 3′UTR end of the *KRAS* gene, mainly single-nucleotide variants, can modify the miRNA binding sites that could potentially post-transcriptionally regulate this gene. Thus, as variants exist in this region, miRNAs do not bind to the target region appropriately and inefficient post-transcriptional regulation could occur (**B**), allowing the full translation of more *KRAS* molecules. This, in turn, could translate into a constant activation of the pathways where *KRAS* participates, promoting cell growth and, thus, neoplastic processes (**C**,**D**). *KRAS* could be regulated by miRNAs at different levels: one, through direct regulation by binding to its 3′UTR region, and another via the regulation of its effector proteins, such as Raf. In addition, mutagenic *KRAS* could also influence the global expression of miRNAs or the regulation of the transcription of factors important for the expression of miRNAs or of the components of the RISC (**E**) complex. Moreover, some miRNAs were observed to be downregulated in PC and, in turn, some of them correlated positively and negatively with *KRAS* expression (**F**). This suggests that when a certain miRNA decreases, the expression of *KRAS* increases and vice versa (negative correlation), or that when the expression of certain miRNA increases, so does the expression of *KRAS*, or if that miRNA decreases, so does the expression of *KRAS* (positive correlation). This is evidence of the important and complex regulation exerted by miRNAs on the expression of a key gene in PC, *KRAS*. Created with Biorender.com.

**Table 1 diseases-12-00152-t001:** miRNAs dysregulated in PC and targeting *KRAS*-specific SNVs.

miARN	*KRAS* SNV	Target Score miRDB *	Context+ Score **
hsa-miR-885-5p	rs8720	<50	−0.155
hsa-miR-877-5p	rs712	79	−0.357
hsa-miR-654-3p	rs190084851	<50	0.069
hsa-miR-647	rs184195260	<50	−0.245
hsa-miR-548c-5p	rs180766260	74	0.133
hsa-miR-513b-5p	rs1141947	67	−0.066
hsa-miR-495-3p	rs184169974	<50	0.060
hsa-miR-377-3p	rs1141947	<50	−0.106
hsa-miR-34b-3p	rs61764370	<50	0.122
hsa-miR-326	rs150334904	72	−0.283
hsa-miR-30e-5p	rs61764373	80	−0.246
hsa-miR-30b-5p	rs61764373	80	−0.246
hsa-miR-30a-5p	rs61764373	80	−0.225
hsa-miR-222-3p	rs189426424	<50	−0.371
hsa-miR-217	rs192263744	92	−0.175
hsa-miR-181d-5p	rs9266	66	−0.113

* Target Score provided by miRDB; ** Context score provided by TargetScan and PolyMIRTS.

**Table 2 diseases-12-00152-t002:** Experimental validation of miRNAs dysregulated in PC and targeting *KRAS*.

miRNA	Cancer Type	Tumor Subtype or Cell Line	Design	Platform	Status	Reference
hsa-miR-885-5p	Pheochromocytoma	Tumors with germline mutations in RET	Tissue: Cancer vs. Normal	qRT-PCR	UP	[30]
hsa-miR-326	Brain cancer	Medulloblastoma	Tissue: Cancer vs. Normal	qRT-PCR	DOWN	[31]
	Esophageal cancer	Barrett’s carcinogenesis	Tissue: Cancer vs. Normal	qRT-PCR	DOWN	[32]
hsa-miR-30b-5p	Colon cancer	Metastatic colorectal cancer	Tissue: Cancer vs. Normal	Northern Blot	DOWN	[33]
	Brain cancer	Medulloblastoma	Tissue: Cancer vs. Normal	qRT-PCR	DOWN	[31]
	Breast cancer	N/A	Tissue: Cancer vs. Normal	qRT-PCR	UP	[34]
hsa-miR-30a-5p	Lung cancer	Lung adenocarcinoma	Blood cells: cancer vs. Normal	qRT-PCR	DOWN	[35]
hsa-miR-222-3p	Prostate cancer	N/A	Tissue: Cancer vs. Normal	qRT-PCR	DOWN	[36]
	Lymphoma	Multiple myeloma (TC4)	Tissue: Subtype1 vs. Subtype2	qRT-PCR	UP	[37]
	Cholangiocarcinoma	N/A	Tissue: Cancer vs. Normal	qPCR	UP	[38]
	Lymphoma	Cutaneous T-cell lymphoma	Tissue: Cancer vs. Normal	qRT-PCR	UP	[39]
	Kidney cancer	N/A	Tissue: Cancer vs. Normal	qRT-PCR	UP	[40]
hsa-miR-217	PC	Solid pseudopapillary neoplasm of pancreas	Tissue: Cancer vs. Normal	Real-Time PCR	DOWN	[41]
	Lung cancer	N/A	Blood	qRT-PCR	UP	[42]

**Table 3 diseases-12-00152-t003:** Comparison of miRNA expression and survival in PC.

miRNA	Type of Association	Log2 Average Expression (Deceased)	Log2 Average Expression (Survivors)	*p* Value
hsa-miR-30a-5p	Mortality	14.22	14.02	0.012
hsa-miR-30b-5p	Survival	8.35	8.56	0.046
hsa-miR-377-3p	Survival	1.30	2.10	0.047
hsa-miR-495-3p	Survival	3.65	4.45	0.013
hsa-miR-885-5p	Survival	0.11	2.65	0.000412

## Data Availability

Data are contained within the article.

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
