# Peer review of "In Silico Identification of Dysregulated miRNAs Targeting KRAS Gene in Pancreatic Cancer"

_diseases, 2024, doi:10.3390/diseases12070152_

Round 1
Reviewer 1 Report
Comments and Suggestions for Authors
The present manuscript aimed to analyze the differential expression of miRNAs in pancreatic cancer (PC), evaluate the dysregulated miRNAs in PC, and in silico analyze their binding or interaction with the KRAS gene. This study employed various platforms/database analysis tools to conduct a comprehensive analysis of KRAS-targeted miRNAs. One of the significant contributions of this study is its comprehensive application of the bioinformatic tools and databases that can be harnessed to study the genetic-epigenetic molecular crosstalk and its potential impact on disease progression, which could be a guideline for researchers interested in employing these tools in their studies.
The study is well presented, and the methods are adequately described. However, I have a few comments to share with the authors.
1- How does this work compare to previous studies on miRNAs targeting KRAS in pancreatic cancer and other cancer types?
2- I suggest an analysis of the overall survival of KRAS in PC using the GEPIA and ULACAN databases.
3- Delineating the function of KRAS in PC, probably by employing the CancerSEA database, would strengthen the work.
4- Ways to evaluate the strength of the predicted miRNA-KRAs interaction should be stated. context scores or other metrics provided by TargetScan should be included in Table 1.
5- What are the cut-off values used for all analyses?
6- The resolution of Figure 4 (collected correlation figures) needs more adjustment.
7- Previous studies that have already validated some of the stated miRNAs as regulators for KRAS expression, such as miR-30 targeting KRAs in colorectal cancer, should be considered and covered.
Author Response
We greatly appreciate your observations and recommendations. Each of them is invaluable for improving our manuscript, and we have carefully taken them into account. Below, we have attached a document with our responses to your valuable comments.

Reviewer 2 Report
Comments and Suggestions for Authors
An article by Dr. Gallegos-Arreola and the group elaborates on the role of aberrant miRNAs targeting the oncogenic KRAS in pancreatic cancer. This study highlights the therapeutic implications of targeting dysregulated miRNAs. This study follows through with well-planned experiments and a hypothesis of the work. This is a well-written and novel idea for targeting KRAS in pancreatic cancer, though a few things need to be addressed before it is ready for acceptance. They are as follows:
1. In the introduction, line 74, the authors mention "signals that lead to cell proliferation, differentiation, metabolism, and migration." The authors must add a reference that mentions the role of oncogenic KRAS in cancer metabolism. For reference, please see PMID: 33870211 and other relevant articles on this topic.
2. Several studies have shown that small noncoding miRNAs regulate YAP/TAZ/ Hippo signaling (PMID: 30380420). It has also been shown that activation of YAP/TAZ acts as a significant regulator of resistance to KRAS inhibitors, and combining Hippo pathways inhibition was capable of significantly extending the response of tumors to KRAS G12C inhibition (PMID: 37729426 and PMID: 24954535). Future studies may elaborate on the role of miRNAs and Hippo signaling in KRAS-driven pancreatic cancer. This will open up an altogether unexplored field for novel therapeutics. The authors must add a few lines discussing this in the discussion part as a possible future aspect to pursue further by adding the relevant references above.
3. Please increase the word/ text sizes in figure 2 and figure 4.
Author Response

(The authors gave the same response as above.)

Round 2
Reviewer 2 Report
Comments and Suggestions for Authors
All concerns addressed, ready for acceptance.
Author Response
We greatly appreciate your suggestions during the review of our manuscript. We are convinced that the suggested changes were invaluable in enhancing the quality of the work.